# Comparative Study of the Suppression Behavior and Fire-Extinguishing Mechanism of Compressed-Gas Aqueous Film-Forming Foam in Diesel Pool Fires

**Long Yan [1], Ning Wang [1], Jingjing Guan [1], Zheng Wei [1], Qiaowei Xiao [2] and Zhisheng Xu [1,*]**

1   Institute of Disaster Prevention Science and Safety Technology, School of Civil Engineering,
    Central South University, Changsha 410075, China; ylong015@csu.edu.cn (L.Y.)
2   Sany Automobile Manufacturing Co., Ltd., Changsha 410100, China
*   Correspondence: zhshxu@csu.edu.cn; Tel.: +86-13707494888

**Abstract:** A compressed-gas fire extinguishing experiment was carried out to analyze the impact of gas-liquid flow ratio, liquid flow rate and driving pressure on the fire suppression efficiency of aqueous film-forming foam (AFFF) in a diesel pool fire, and a possible fire-extinguishing mechanism was proposed. A fire suppression test showed that AFFF at a gas-liquid flow ratio of 16 between the range of 5 to 24 had the fastest fire-extinguishing temperature drop rate (16.67 °C/s), the shortest fire-extinguishing time, of 42 s, and the lowest foam solution consumption of 230 g, exhibiting the best fire suppression performance. Meanwhile, the fire suppression efficiency of AFFF improved with the augmentation of either liquid flow rate or system driving pressure. Based on fluid mechanics and combustion science, a foam fire-extinguishing mechanism was proposed to explain the influence of system parameters such as gas-liquid ratio, liquid flow rate and driving pressure on key combustion parameters such as temperature drop rate, evaporation rate and combustion rate, which can better illustrate the change in fire extinguishing performance.

**Keywords:** aqueous film-forming foam; gas-liquid flow ratio; driving pressure; fire-extinguishing efficiency

## 1. Introduction

With the progression of the petrochemical industry, flammable hydrocarbon fire has caused lots of casualties and property losses, and traditional water-based fire-extinguishing agents cannot apply to the extinguishing of hydrocarbon fire [1]. Fire suppressing foam is extensively used to extinguish fuel fires in the petrochemical industry, which is a result of its outstanding coverage performance, cooling performance and performance in preventing recrudescence. Among various fire-fighting foams, aqueous film-forming foam (AFFF) is one of the most effective methods to suppress petrochemical fires. Because AFFF can spread quickly on the oil surface, it has become an ideal extinguishing agent for liquid fires. AFFFs have better fire extinguishing performance when combined with compressed air foam systems or injected by using liquid injection methods. The super superior fire-extinguishing effect of AFFF in hydrocarbon pool fire is ascribed to the dual influence of the foam layer and aqueous film [2].

In the past few decades, many efforts have focused on the influence of foam components, the foam fire-fighting system and oil film interaction on the film-foam property, foam stability, spreading behavior and fire-extinguishing performance of AFFF in oil fires [3–6]. Sheng et al. [7,8] investigated the fire suppression behavior of foam extinguishing agents with various components, and found that the AFFF containing fluorocarbon surfactants could diffuse faster on the surface of the fuel [7,8]. Moreover, the introduction of foam stabilizers such as xanthan gum, triethanolamine, lauryl alcohol and carboxymethyl cellulose can effectively enhance the diffusion performance and drainage performance of AFFF.

Ranjbar et al. [9] reported that the aqueous film with water-soluble surfactant could not alleviate its evaporation rate of the fuel. Generally, the extinguishing capability of AFFF is related to the cooling and covering effects of foam. The gas-liquid flow ratio of foam affects the moisture condition, fluidity and stability of the foam and determines the character of expansion and drainage of foam [10]. Feng et al. [11] researched the super mixing process by setting values of parameters and thus establishing a mathematical model of the two-phase mixing process in view of the real situation. Lee et al. [12] found the effect of surfactant and mixing ratio in compressed air foams on the fire suppression properties of AFFF. The conclusion illustrated that AFFF showed the best fire suppression effect on a gas-liquid flow ratio of 7. Chen et al. [13] reported that foam coverage was related to fire-extinguishing efficiency, and the highest foam coverage rate was observed at the foam expansion ratio of 10. Zhao et al. [14] found that CAF had an excellent extinguishing effect in extinguishing oil tank fires by comparing the low foaming and multiple foam fire-extinguishing systems. Zhao et al. [15] studied the combined effect of fluoro-protein foam and BTP in improving the fire suppressing efficiency of foam. Meanwhile, it was found that the foam showed the best efficiency in suppressing oil pool fires at an optimal gas-liquidvolume ratio of 9. Kang et al. [16] revealed that the protein foam had the best fire extinguishing effect in diesel pool fires under the system driving pressure of 0.5 MPa and a liquid flow rate of 50 L/h. Yu et al. [1] revealed that the oil film interaction had a negative effect on the foam stability and foam, and the fluorinated foam was less effected by the oil film compared to fluorine-free foam during the fire extinguishing process. Chen et al. [17] found that the compressed nitrogen foam had a better fire-extinguishing effect than that of compressed air foam on inhibiting coal spontaneous combustion due to the good endothermic process and dilution effect. Li et al. [18] evaluated the effect of different types of foam fire extinguishing agents on diesel fires by combining the temperature changes of fires after spraying foam.

Apart from the experimental analysis of fire-fighting foams, many researchers have focused on the application of model analysis and numerical analysis to quantitatively assess the fire suppression behavior of fire-fighting foams. Pan et al. [19] studied the supply intensity of AFFF through a series of fire suppression experiments, and established a minimum supply intensity prediction model of AFFF to effectively predict the coverage process of AFFF under cold spray. Conroy et al. [20] established a cooling model of AFFF for describing the time evolution of the temperature profile by numerically solving a transient, one-dimensional, heat-conduction equation in the liquid pool and in the foam layer. Galaj et al. [21] evaluated the effect of average droplet diameter on the extinguishing efficiency of the foam extinguishing agent and established the assumptions and relationships that represent the mathematical model of the fire extinguishing process. The model includes a unique method for determining the spraying area in the basic unit of the on-site fire model. Zhu et al. [22] derived the effective concentration based on the diffusion equation and analyzed its variation with diffusion coefficient and diffusion time. In view of these research results, a mathematical model was established to evaluate the liquid separation performance and diffusion performance of a foam extinguishing agent. However, the existing petrochemical fire extinguishing theory was mainly based on some qualitative and general principles in the ideal fire state, which lacked quantitative or semi-quantitative analysis of physical and chemical processes. Therefore, it is necessary to quantitatively analyze the fire-extinguishing behavior of foam from the perspective of fluid mechanics and combustion science, which is beneficial to develop new foam extinguishing agents and foam fire extinguishing prediction models.

In this paper, a laboratory fire-extinguishing experiment was carried out to analyze the effect of gas-liquid flow ratio, liquid flow rate and driving pressure on the fire-extinguishing properties of AFFF. The fire-extinguishing process of the foam with appropriate gas-liquid flow ratio, liquid flow rate and driving pressure were analyzed carefully. Based on the experimental results and theoretical analysis, this paper establishes a semi-qualitative and semi-quantitative theoretical heat transfer process of foam extinguishing agents to suppress diesel pool fires, so as to clarify the fire extinguishing mechanism of foam extinguishing agents.

## 2. Experimental Section

The fire-extinguishing test was carried out using a self-assembled compressed-gas fire-extinguishing system according to the procedure previously reported [10]. A diagram of the fire extinguishing device is shown in Figure 1. In the test, 6% aqueous film-forming foam liquid (6% AFFF) was produced by Jiangsu Anrui Life-saving Fire-fighting Equipment Company of China. In the foam fire extinguishing system, the foam liquid is driven by the pressure generated by the nitrogen bottle. Diesel (0#, Sinopec Group, Changsha, China) is used as fuel and ignited with alcohol. Firstly, 1.1 kg of diesel was poured into a stainless-steel oil pan whose size was 0.5 m × 0.5 m × 0.1 m and thickness 5 mm. When the diesel combustion remained stable, the pressure gauge was turned on to stabilize the pressure at the set value. Foam was produced in the mixing process of nitrogen and foam liquid in the T-type mixing chamber and different gas-liquid flow ratios were obtained by changing the value of the liquid flow rate via the gas flow-meter. The flame temperature was measured by WRNK-131 thermocouple during the experimental process. The height difference between the nozzle and the oil pan was 1.5 m, and all the foam was injected into the oil pan. The AFFF consumption was collected by the H2 electric balance (Kaifeng Company, Kaifeng, China). The flame extinction feature data were recorded with an infrared thermal imager, which was FOTRIC X-Ti6. At the beginning of the fire extinguishing test, 500 g diesel oil was poured into a stainless steel oil pan with a size of 0.5 m × 0.5 m × 0.1 m. After leveling the stainless steel oil pan, a high temperature jet igniter was used for ignition. After 60 s combustion of the diesel, the start-up agent bottle or air compressor was opened for gas transportation, then the gas flowmeter and liquid flowmeter were adjusted to control liquid flow, gas flow and gas-liquid ratio. The foam extinguishing agent was vertically transported to the oil pan through the pipeline, and the injection distance was 1.5 m. After the flame was suppressed, the air compressor or starting agent bottle is closed, and the gas flowmeter and liquid flowmeter are closed. A blank contrast experiment was set up before the fire tests to determine the preburn time and the fire temperature. All the fire-extinguishing processes were recorded with a Canon digital camera. Each group of tests was repeated several times under conditions of wind speed less than 1.5 m/s and temperature of 20~25 °C.

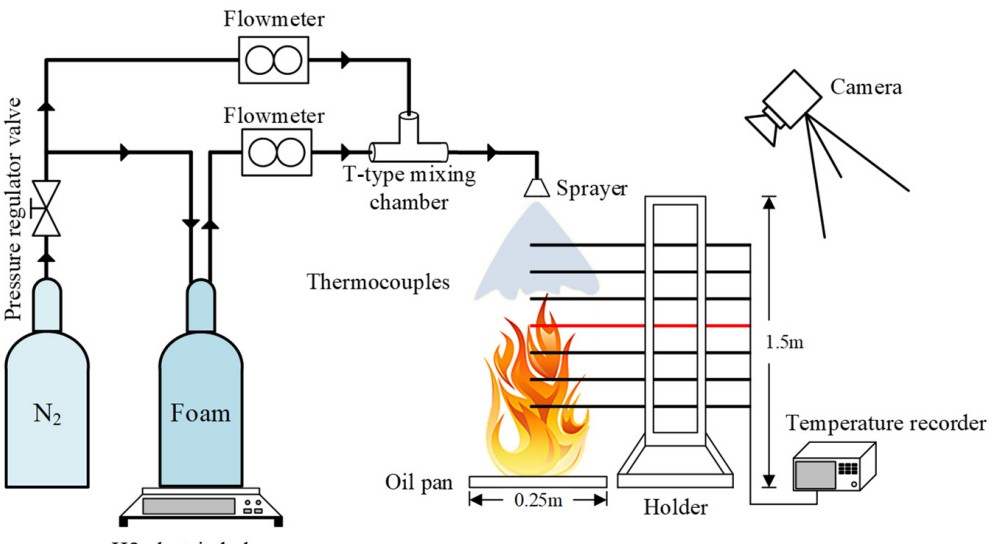

**Figure 1.** Diagram of fire extinguishing device (the data used in this paper were measured with the red thermocouple).

## 3. Results and Discussion

### 3.1. Influence of Gas-Liquid Flow Ratio on the Fire Suppression Behavior of AFFF

Figure 2 shows the temperature variation curve under different gas-liquid flow ratios ($\alpha$), which is measured with a thermocouple at the center of the flame. In addition, the

combustion process without an extinguishing agent was recorded as the blank control group. AFFF sprayed out after burning for 100 s, and the temperature soared rapidly in a short time. Then, the temperature dropped quickly on account of the cooling effect and suffocation effect of AFFF. Because AFFF enhances the flame, the temperature curve had an obvious peak after the AFFF was ejected. AFFF was injected into the fuel layer, which caused the fuel vapor inside the fuel to be quickly released into the air, thus increasing the fuel vapor concentration in the mixed gas as well as intensifying the combustion of the oil pool fire. After a short period of the above process, as the foam content in the mixed layer enlarged, the volume fraction of the fuel decreased and the evaporation rate of the fuel descended, thereby inhibiting the progress of combustion.

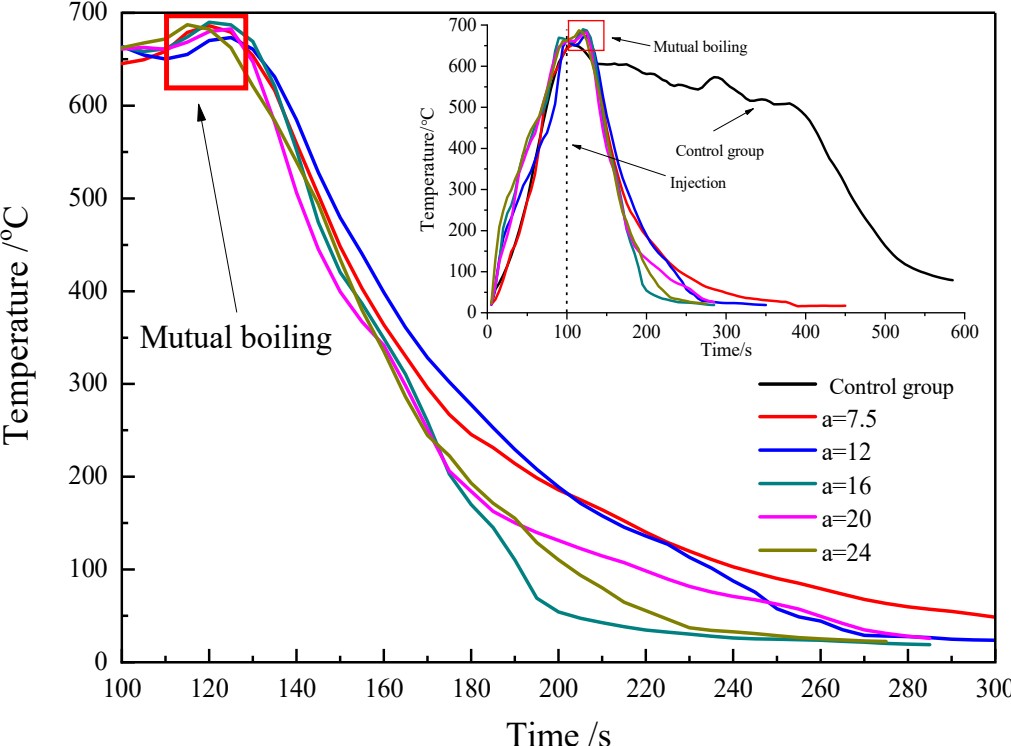

**Figure 2.** Temperature variation curves of diesel pool fire with application of different gas-liquid flow ratios of AFFF.

As seen from Table 1, the extinguishing time and AFFF solution consumption decreased rapidly and then increased slightly with the augmentation of the gas-liquid flow ratio, and the foam with a gas-liquid flow ratio of 16 showed the minimum extinguishing time of 42 s and the lowest AFFF solution consumption of 230 g. Meanwhile, the fire-extinguishing time and extinguishing agent consumption of AFFF at a gas-liquid flow ratio of 16 were reduced by 95 s and 365 g, respectively, compared with the AFFF at a gas-liquid flow ratio of 5. In particular, the temperature dropping rate of foam with different $\alpha$ in the fire-extinguishing process presented an order of $\alpha = 16 > \alpha = 24 > \alpha = 20 > \alpha = 12 > \alpha = 7.5$. With the augmentation of the gas-liquid flow ratio, the increase in air volume fraction and expansion ratio led to the decrease in fuel vapor concentration in the gas mixture layer, which inhibited the combustion reaction. However, an excessive gas-liquid ratio would lead to a decrease in the drainage rate and foam concentration in the mixed layer, which would result in an increase in fuel concentration, which causes an increase in fuel evaporation rate, thereby reducing the inhibition of the foam on combustion.

**Table 1.** Fire-extinguishing time and extinguishing dosage agent of AFFF at different gas-liquid flow ratios.

| Gas-Liquid Flow Ratios | Expansion Ratio | Drainage Rate (g/min) | Driving Pressure (MPa) | Liquid Flow Rate (L/h) | Extinguish Time (s) | AFFF Solution Consumption (g) | Temperature Drop Rate (°C/s) |
|---|---|---|---|---|---|---|---|
| 5 | 19.9 | 0.212 | 0.3 | 20 | 137 ± 7 | 595 ± 9 | 4.32 ± 0.3 |
| 7.5 | 22.3 | 0.161 | 0.3 | 20 | 82 ± 7 | 365 ± 8 | 8.54 ± 0.4 |
| 10 | 24.1 | 0.133 | 0.3 | 20 | 67 ± 6 | 293 ± 9 | 10.45 ± 0.4 |
| 12 | 24.7 | 0.139 | 0.3 | 20 | 63 ± 5 | 261 ± 7 | 11.11 ± 0.5 |
| 14 | 25.1 | 0.136 | 0.3 | 20 | 62 ± 5 | 257 ± 8 | 11.29 ± 0.4 |
| 16 | 25.7 | 0.140 | 0.3 | 20 | 42 ± 4 | 230 ± 4 | 16.67 ± 0.5 |
| 18 | 25.9 | 0.136 | 0.3 | 20 | 48 ± 4 | 248 ± 3 | 14.58 ± 0.4 |
| 20 | 26.1 | 0.131 | 0.3 | 20 | 56 ± 5 | 242 ± 6 | 12.51 ± 0.4 |
| 22 | 26.1 | 0.127 | 0.3 | 20 | 65 ± 6 | 260 ± 7 | 10.77 ± 0.5 |
| 24 | 26.2 | 0.125 | 0.3 | 20 | 63 ± 4 | 256 ± 9 | 12.84 ± 0.5 |

The flame extinction process of the foam with a gas-liquid flow ratio of 16 is presented in Figure 3. As shown in Figure 3, the flame height firstly increases and then decreases after the injection of foam, while the increase in the flame height was caused by mutual boiling. The mutual boiling phenomenon can be ascribed to the mixture of foam and fuel causing the fuel vapor in the mixing layer to quickly volatilize into the air after the foam has been added to the oil pan, resulting in the increase in combustion intensity.

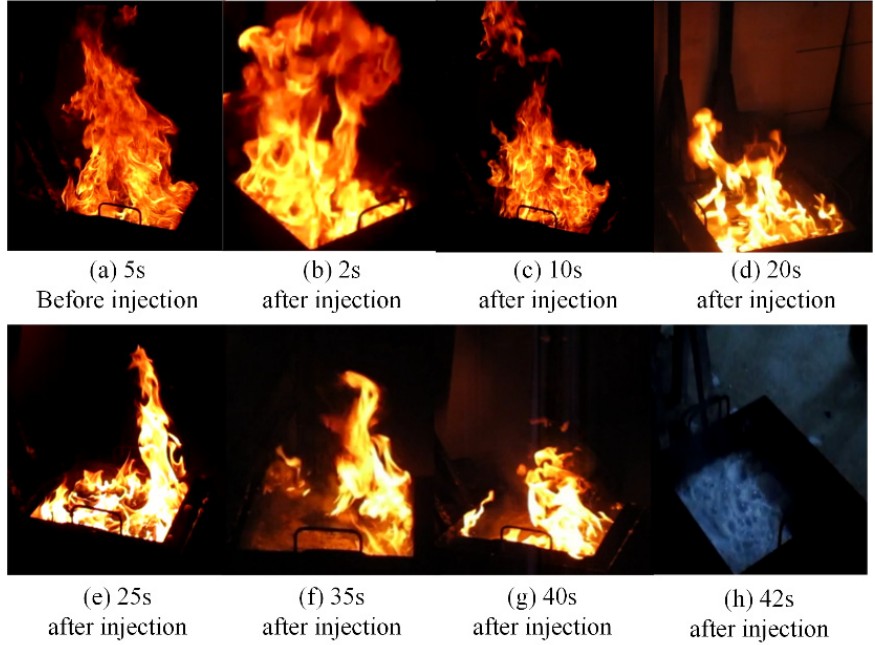

(a) 5s
Before injection

(b) 2s
after injection

(c) 10s
after injection

(d) 20s
after injection

(e) 25s
after injection

(f) 35s
after injection

(g) 40s
after injection

(h) 42s
after injection

**Figure 3.** The flame extinction process of the foam with a gas-liquid flow ratio of 16.

*3.2. Influence of Liquid Flow Rate on the Fire Suppression Behavior of AFFF*

The fire-extinguishing times of AFFF at different foam volume flows are shown in Table 2. The results show that as the volume flow of foam liquid increased, the fire-extinguishing time and the consumption of the fire-extinguishing agent decreased. When the liquid volume flow of AFFF was increased from 20 L/h to 60 L/h, the extinguishing time and the consumption of the AFFF solution were reduced by 15 s and 29 g, respectively. As the foam volume flow increased, the fire-extinguishing time decreased. Because the foam volume flow flowing into the mixing layer per unit time increased, the volume fraction of water and air in the fuel–foam mixing layer increased. The volume fraction of

the fuel vapor decreased, so the reduction in the evaporation rate of fuel vapor weakened the combustion of the fuel.

**Table 2.** Fire-extinguishing efficiency of AFFF at different foam volume flow rates.

| Liquid Flow Rate (L/h) | Driving Pressure (MPa) | Gas-Liquid Flow Ratio | Extinguish Time (s) | AFFF Solution Consumption (g) |
|---|---|---|---|---|
| 20 | 0.3 | 16 | 42 ± 4 | 230 ± 4 |
| 40 | 0.3 | 16 | 35 ± 6 | 213 ± 4 |
| 60 | 0.3 | 16 | 27 ± 4 | 201 ± 6 |

As shown in Figure 4, the fluidity and thickness of the foam were enhanced with the rise in the liquid volume flow of AFFF, resulting in the strengthening of the stifling effect and covering effect of AFFF.

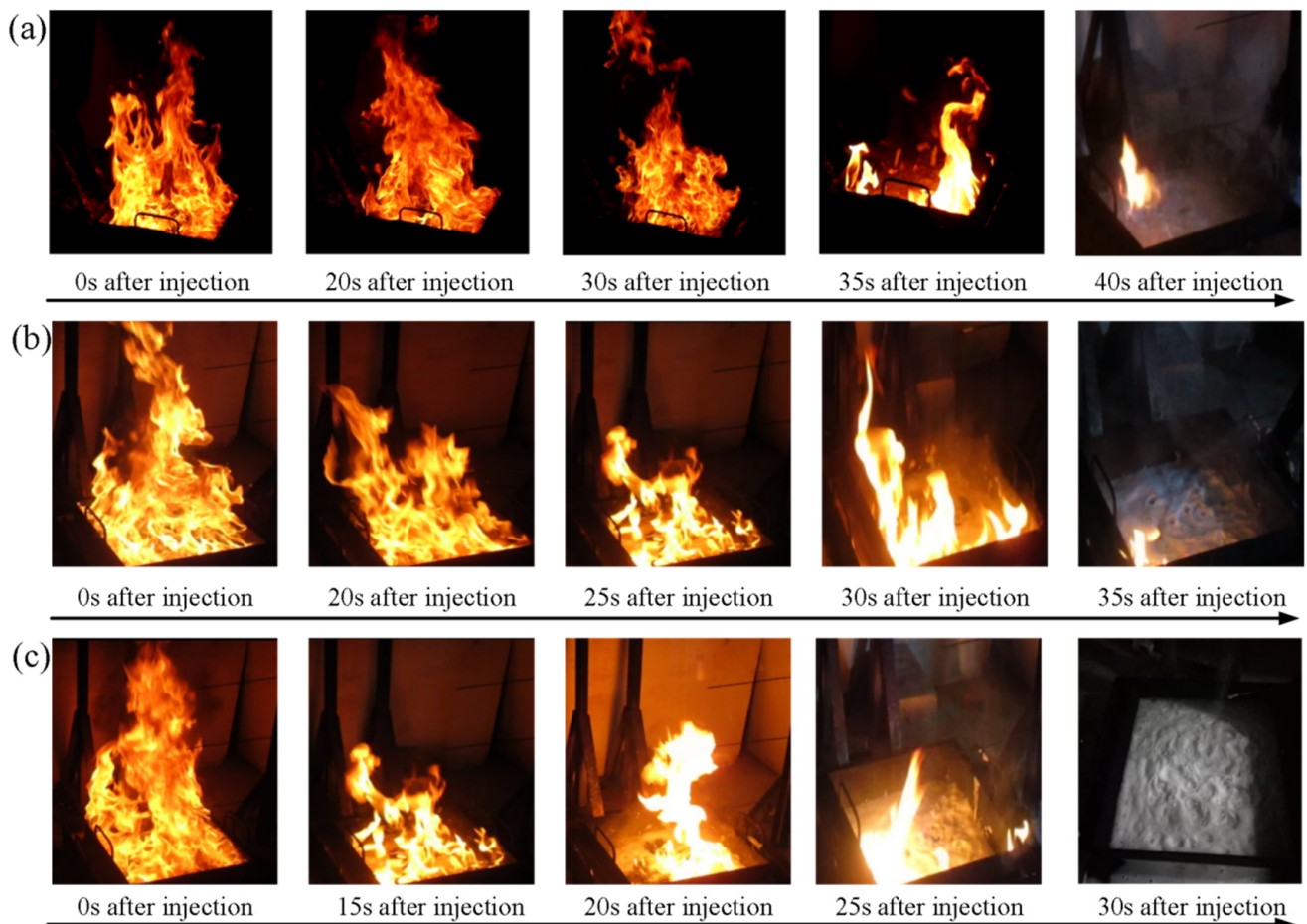

**Figure 4.** The flame extinction process of AFFF with different foam volume flow rates: (**a**) 20 L/h, (**b**) 40 L/h and (**c**) 60 L/h.

### 3.3. Influence of System Driving Pressure on the Fire Suppression Behavior of AFFF

Table 3 shows the AFFF fire-extinguishing time and fire-extinguishing agent consumption of 60 L/h at different system driving pressures. As seen in Table 3, the fire-extinguishing time and extinguishing agent consumption of AFFF decreased with the increase in system driving pressure. When the system driving pressure of AFFF increased from 0.3 MPa to 0.5 MPa, the fire-extinguishing time and extinguishing agent consumption were reduced by 11 s and 29 g, separately. Figure 5 illustrates that the AFFF with a higher system driving pressure had higher initial potential energy, which increased the thickness

of the mixed layer to accelerate the heat exchange rate between the fuel and the foam. As more foam layer was accumulated, the flame temperature slowly dropped more.

**Table 3.** Fire-extinguishing time and extinguishing dosage agent of AFFF with different system driving pressures.

| Driving Pressure (MPa) | Liquid Flow Rate (L/h) | Gas-Liquid Flow Ratios | Extinguish Time (s) | AFFF Solution Consumption (g) |
|---|---|---|---|---|
| 0.3 | 60 | 16 | 27 ± 4 | 201 ± 6 |
| 0.4 | 60 | 16 | 22 ± 3 | 189 ± 7 |
| 0.5 | 60 | 16 | 16 ± 3 | 172 ± 4 |

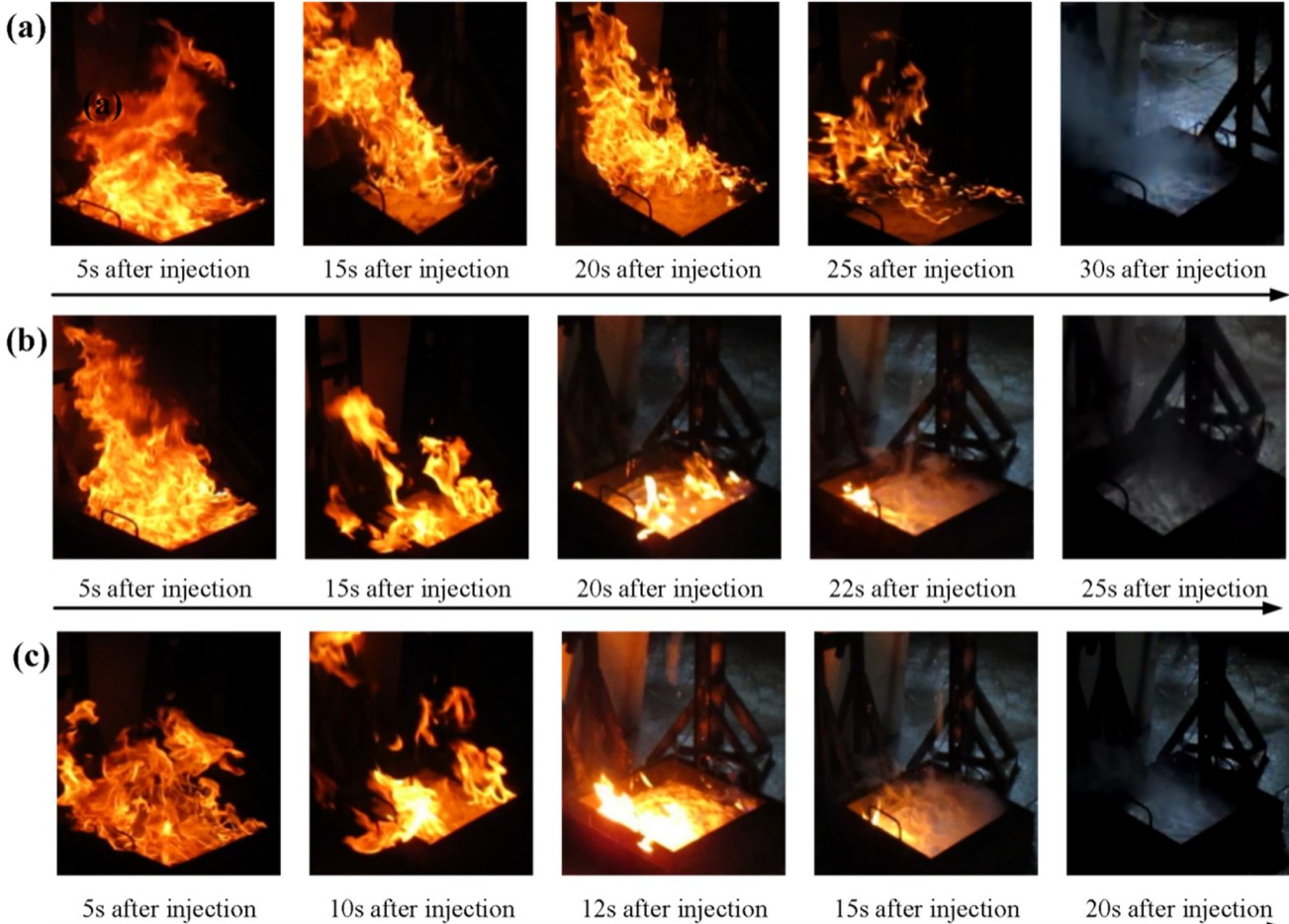

**Figure 5.** The flame extinction process of AFFF with different system driving pressures: (**a**) 0.3 MPa, (**b**) 0.4 MPa and (**c**) 0.5 Mpa.

*3.4. Fire Suppression Mechanism of AFFF in Diesel Pool Fire*

The diagram of the fire-extinguishing process is presented in Figure 6. As shown in Figure 6, the fire extinguish mechanism in the experiment was abstracted into a rectangular area with three sub-regions, namely, the fuel layer, fuel–foam mixing layer and mixed gas layer. The liquid phase of the mixing layer was water and fuel, while the gas mixing layer was composed of fuel steam and air. In order to interrupt the combustion process of fuel, the evaporation rate of liquid was required to meet the following conditions [23],

$$G_1 < \frac{\dot{Q}_E - \dot{Q}_1}{\varphi \Delta H_c - L_V} \tag{1}$$

where $G_1$ is the fuel evaporation rate, $\Delta H_c$ is the fuel combustion heat, $\varphi$ is the percentage of the heat released by the liquid combustion feedback to the liquid surface, $\dot{Q}_E$ is the heating rate of the external heat source, $\dot{Q}_1$ is the heat loss rate of the mixing layer and $L_V$ is the latent heat of the fuel evaporation.

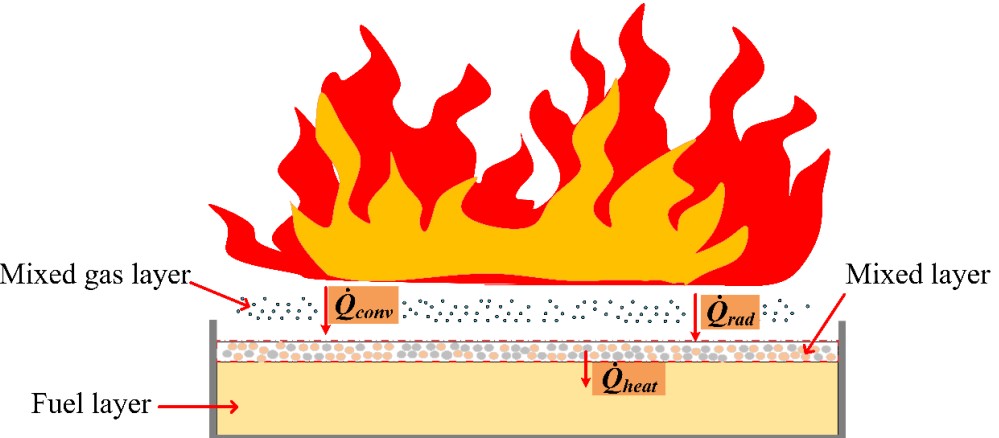

**Figure 6.** Heat transfer diagram of fire-extinguishing process.

It can be seen from Equation (1) that the combustion of fuel is controlled by the evaporation rate of the fuel, the heating rate of the external heat source and the heat loss rate. The combustion interruption can be attributed to the decrease in fuel evaporation rate, the decrease in the external heat source heating rate and the increase in the heat loss rate of the fuel foam mixing layer.

Furthermore,

$$\dot{Q}_E = \dot{Q}_{conv} + \dot{Q}_{rad} \tag{2}$$

$$\dot{Q}_1 = \dot{Q}_{heat} + \rho_l \dot{V} \varphi_l \left( L_V + (T_0' - T_0) C_P \right) \tag{3}$$

where $\rho_l$ is the density of the foam liquid, $\dot{V}$ is the Foam volume injected per unit time, $\varphi_l$ is the volume fraction of water, $\varphi_l = \frac{1}{1+\alpha}$ ($\alpha$ is the gas-liquid flow ratio), $L_V$ is the latent heat of water evaporation, $T_0'$ is the boiling point of water, $T_0$ is the temperature of the foam liquid and $C_P$ is the specific heat capacity of water.

It can be seen from Equations (2) and (3) that the external heat is mainly generated by the mixed combustion of fuel vapor and air in the mixed gas layer. The heat loss rate depends on the heat conduction between the mixing layer and the fuel layer, the foam volume of the injection tank and the volume fraction of water. With the increase in the injection rate, the $\dot{V}$ and $\dot{Q}_1$ gradually increase, thus resulting in the rapid extinguishing of the flame. This is consistent with the observation from Table 3.

In order to analyze the influencing factors on evaporation rate, $G_1$ can be expressed as Equation (4).

$$G_1 = D(c_g - c_l) \tag{4}$$

where $D$ is diffusion coefficient, $c_g$ is fuel vapor concentration in the mixing layer and $c_l$ is fuel vapor concentration in the fuel layer.

As can be seen from Equation (4), $G_1$ mainly depends on the fuel concentration difference and temperature difference between the two layers. AFFF formed a covered water film on the upper surface of the mixing layer, resulting in a decrease in fuel evaporation. The decrease in fuel evaporation rate reduced the fuel vapor in the air, resulting in flame extinction. In order to illustrate the effect of AFFF on pool fires, regarding coverage, cooling and suffocation, Figure 7 is introduced to represent the temperature change during the fire-extinguishing process. It can be seen from Figure 7, with the injection of the foam liquid, that the temperature of the fuel layer and the foam–fuel mixture layer decreased by

about 300 °C within 30 s. In addition, the temperature of the flame dropped to 549.6 °C within 35 s of adding foam, thus exhibiting the excellent cooling performance of AFFF.

**Figure 7.** Flame temperature variation in fire-extinguishing process at gas-liquid flow ratio of 16.

Based on the above analysis, the fire-extinguishing properties of AFFF were mainly related to cooling, covering and asphyxiation. During the fire-extinguishing process, evaporated water from the foam had a superior cooling effect that effectively decreased the fuel vapor concentration in the mixed gas layer and reduced the flame temperature of the combustion zone. Meanwhile, the water film and foam film formed on the surface of the oil surface acted as an effectively physical barrier that diminished the evaporation rate of the flammable liquid. Moreover, the defoaming foam could transfer to inert gaseous products and diffuse into the gas phase zone, reducing the concentration of fuel vapor as well as inhibiting the combustion intensity.

## 4. Conclusions

In this paper, a self-assembly fire-extinguishing device was used to research the fire-suppressing performance of AFFF on diesel pool fire. With the rise in the gas-liquid flow ratio, the fire-extinguishing efficiency of AFFF firstly increased and then decreased, which was ascribed to the reduction in fuel vapor in the gas mixture layer. When the gas-liquid flow ratio was 16, the temperature drop rate was the fastest, at 16.67 °C/s, and the mass of the foam solution was the lowest, at 230 g. Similarly, the fire-extinguishing efficiency of AFFF increased with the augmentation of the foam flow, which is ascribed to the reduction in the evaporation rate of the fuel vapor. The fire-extinguishing time and extinguishing agent consumption of AFFF with a foam flow rate of 60 L/h were 15 s and 29 g, respectively, which was lower than those of AFFF with a foam flow rate of 20 L/h. With the increase in system driving pressure, the fire-extinguishing time and the consumption of extinguishing agent were gradually reduced, which was ascribed to the

enhancement of the heat exchange rate between foams with hot fuels. Compared with dates under 0.3 MPa driving pressure, the extinguishing time and extinguishing agent consumption of AFFF under a 0.5 MPa driving pressure were reduced by 11 s and 29 g, respectively. The fire-extinguishing mechanism of AFFF in pool fires was attributed to the cooling effect, covering effect and suffocation effect, which were affected by the gas-liquid flow ratio, liquid flow rate and driving pressure of AFFF. In view of the theoretical analysis of foam fire suppression behavior, the temperature drop rate is mainly affected by the gas-liquid ratio and injection rate of the foam extinguishing agent.

**Author Contributions:** L.Y.: Conceptualization, Methodology, Supervision, Writing—Review and Editing. N.W.: Writing—original draft, Data curation, Investigation. J.G.: Validation, Visualization. Z.W.: Visualization. Q.X.: Visualization. Z.X.: Supervision, Project Administration. All authors have read and agreed to the published version of the manuscript.

**Funding:** This work was supported by the National Natural Science Foundation of China [grant number 52176146] and the Key Research and Development Program of Hunan Province [grant number 2021SK2054].

**Data Availability Statement:** Raw data are held by the author and may be made available upon request.

**Conflicts of Interest:** The authors declare that they have no known competing financial interests or personal relationships that could have appeared to influence the work reported in this paper.

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
