# Peer review of "Comparative Study of the Suppression Behavior and Fire-Extinguishing Mechanism of Compressed-Gas Aqueous Film-Forming Foam in Diesel Pool Fires"

_fire, doi:10.3390/fire6070269_

Round 1

Reviewer 1 Report

REVIEW OF FIRE 2023 Yan et al

This paper describes 0.5 m x 0.5 m square pool fire suppression by compressed nitrogen foam (CNF) generated inside a mixing device and sprayed from by a nozzle from the top of the fire.  The foam is generated from Aqueous Film Forming Foam (AFFF) solution and diesel fuel is used in the square pan.  AFFF solution flow rate, ratio of nitrogen gas flow rate to AFFF solution flow rate, and nitrogen gas pressure are varied as parameters.  The effect of the parameters on fire extinction time, temperature versus time profiles at different positions along the height of the fire, and the amount of AFFF solution applied on the fire are reported.  The authors find that increasing the liquid flow rate by a factor of 3 decreases the fire extinction time by a factor of 1.6.  The authors also find that as the ratio of gas-to-liquid flow rates is increased from 5 to 24, the extinction time decreases, and then increases.  But surprisingly, the foam expansion ratio remains at 25 and does not change significantly above 7.5 ratio (measurement method for expansion ratio is not described).  A minimum extinction time of 40 s occurs at a  ratio of 16. Furthermore, as the nitrogen pressure is increased by a factor of 1.7, the extinction time decreases by a factor of 1.7.

While the above experimental results are interesting, I have serious concerns on how the results are interpreted.  The so called “model” is not a model predicting any quantity based on some conservation principle.  There are no predicted numbers from the model that could be compared with measurements.  The authors simply evoke some parameters by writing heat balance across a fuel layer, which appears to be wrong and more importantly fail to explain the observed effects even in a qualitative way.  Some of the equations including equation (4) do not make any sense and most likely wrong.  I recommend authors stick to reporting just the experimental results and remove the “model” from the title and remove section 3.4. from the text and avoid any interpretations based on that.

Specific comments:

1. The language in INTRODUCTION section is crude with many meaning less and awkward sentences written without care especially in the second paragraph of the section.

2. Specify the trade names for AFFF solution and diesel fuel so that readers can find them.

3. What kind of nozzle is used? How is expansion ratio measured ?

4. Section 3.1, first paragraph: “AFFF is injected into fuel layer” ?  Are you introducing AFFF solution in two different ways?

5. Last sentences of section 3.4 make no sense.

6. CONCLUSIONS: line 3 “fire extinguishing efficiency” is not defined and has no meaning. 

7. Remove all sentences referring to “model”.

The language in INTRODUCTION section is crude with many meaning less and awkward sentences written without care especially in the second paragraph of the section.  Many awkward sentences in rest of the paper.  Some are incomprehensible.

Author Response

This paper describes 0.5 m x 0.5 m square pool fire suppression by compressed nitrogen foam (CNF) generated inside a mixing device and sprayed from by a nozzle from the top of the fire.  The foam is generated from Aqueous Film Forming Foam (AFFF) solution and diesel fuel is used in the square pan.  AFFF solution flow rate, ratio of nitrogen gas flow rate to AFFF solution flow rate, and nitrogen gas pressure are varied as parameters.  The effect of the parameters on fire extinction time, temperature versus time profiles at different positions along the height of the fire, and the amount of AFFF solution applied on the fire are reported.  The authors find that increasing the liquid flow rate by a factor of 3 decreases the fire extinction time by a factor of 1.6.  The authors also find that as the ratio of gas-to-liquid flow rates is increased from 5 to 24, the extinction time decreases, and then increases.  But surprisingly, the foam expansion ratio remains at 25 and does not change significantly above 7.5 ratio (measurement method for expansion ratio is not described).  A minimum extinction time of 40 s occurs at a  ratio of 16. Furthermore, as the nitrogen pressure is increased by a factor of 1.7, the extinction time decreases by a factor of 1.7.

While the above experimental results are interesting, I have serious concerns on how the results are interpreted.  The so called “model” is not a model predicting any quantity based on some conservation principle.  There are no predicted numbers from the model that could be compared with measurements.  The authors simply evoke some parameters by writing heat balance across a fuel layer, which appears to be wrong and more importantly fail to explain the observed effects even in a qualitative way.  Some of the equations including equation (4) do not make any sense and most likely wrong.  I recommend authors stick to reporting just the experimental results and remove the “model” from the title and remove section 3.4. from the text and avoid any interpretations based on that.

  1. The language in INTRODUCTION section is crude with many meaning less and awkward sentences written without care especially in the second paragraph of the section.

Reply: Thanks for your advice. The introduction has been modified and the language has been improved.

  1. Specify the trade names for AFFF solution and diesel fuel so that readers can find them.

Reply: Thanks for your advice. The trade names of AFFF has been showed in manuscript.As seen in section 2, 6% aqueous film-forming foam liquid (6% AFFF) was produced by Jiangsu Anrui Life-saving Fire-fighting Equipment Company of China.

  1. What kind of nozzle is used? How is expansion ratio measured ?

Reply: Thanks for your advice. The type of nozzle is faucet which is used in normal compressed air foam system (CAFS). The purpose of this paper is to study the best gas-liquid ratio, the best fire extinguishing pressure and the best fire extinguishing speed of AFFF. The research purpose of this paper is to confirm the best gas-liquid mixing ratio suitable for AFFF to extinguish petrochemical fire by observing the temperature change curve and flame morphology change diagram in the fire extinguishing process of different gas-liquid ratios. Hence it, the expansion ratio of AFFF used in manuscript has not been measured.

  1. Section 3.1, first paragraph: “AFFF is injected into fuel layer”? Are you introducing AFFF solution in two different ways?

Reply: Thanks for your advice. The AFFF is injected vertically through a nozzle 1.5 m high from the fuel, and this is the only injection method.

  1. Last sentences of section 3.4 make no sense.

Reply: Thanks for your advice. The last sentences of section 3.4 has been modified and last sentences of section 3.4 is used to explain the fire extinguishing mechanism of AFFF.

  1. CONCLUSIONS: line 3 “fire extinguishing efficiency”is not defined and has no meaning.

Reply: Thanks for your advice. Fire extinguishing efficiency is a qualitative index, which represents the extinguishing time of foam extinguishing agent, the rate of flame temperature reduction and so on.

  1. Remove all sentences referring to “model”.

Reply: Thanks for your advice. The description of the model has been modified or removed.

Reviewer 2 Report

The article provides a general overview of the temperature changes during the extinguishing process but lacks specific data or in-depth analysis. It is recommended to include measured temperature data and depict graphs that visually illustrate the temperature changes. Additionally, conduct a more detailed analysis of the impact of AFFF on the combustion and fire extinguishing process.

Discuss your research findings by comparing them with previous studies conducted in this field. Are your findings consistent or contradictory to prior research? Explain why this is the case and highlight the unique contributions of your study.

(page 4-11).

Author Response

The article provides a general overview of the temperature changes during the extinguishing process but lacks specific data or in-depth analysis. It is recommended to include measured temperature data and depict graphs that visually illustrate the temperature changes. Additionally, conduct a more detailed analysis of the impact of AFFF on the combustion and fire extinguishing process.

Reply: Thanks for your valuable advice. Fig 2 is the curve of AFFF fire extinguishing temperature under different gas-liquid flow ratios. These data are measured by thermocouples with a height of 1.5 m from the oil pan. Because of the hysteresis of temperature change during the fire extinguishing process, the temperature data measured by thermocouples are often used to determine the maximum flame temperature and compare the cooling rate of flame under different gas-liquid flow ratios under AFFF. The experimental results of flame extinction time are mainly determined by the photographs of different periods taken by the camera, as shown in Fig. 3.

Discuss your research findings by comparing them with previous studies conducted in this field. Are your findings consistent or contradictory to prior research? Explain why this is the case and highlight the unique contributions of your study.

Reply: Thanks for your advice. The purpose of manuscript are aimed to compare the fire extinguishing time and temperature drop rate of foam extinguishing agent under different flow rate, gas-liquid mixing ratio and driving pressure. Meanwhile, due to the different fire extinguishing efficiency of foam extinguishing agent under different technical parameters of fire extinguishing system, it is impossible to compare the performance of different fire extinguishing agents. Hence it, the manuscript only compares the experimental results of AFFF under different fire extinguishing technical parameters.

Reviewer 3 Report

The research topic is well chosen, and paper has written nicely but for making to it more impactful following points may be included in revised manuscript:

Introduction:

(i) Describe AFFF properly with chemical equations. How it will be reacting with diesel fuel layers.

(ii) Why has author taken diesel fuel rather than any other standard fuels?

(iii) Must be included few literatures from burning behavior of diesel pool fire inside compartment.

Results and Discussion:

(iv) Section 3.1, 3.2 & 3.3 are not compared with any similar studies held in past, kindly compare presented results with another.

(v) In section 3.4, Equations 1, 2, 3 & 4 are not properly cited i.e., from where it has been taken?

(vi) While doing fire suppression modelling of AFFF in diesel pool fire, kindly the quantity of diesel fuel and AFFF along with how reactions are being at mixed gas layer?

Author Response

The research topic is well chosen, and paper has written nicely but for making to it more impactful following points may be included in revised manuscript:

Introduction:

(i) Describe AFFF properly with chemical equations. How it will be reacting with diesel fuel layers.

Reply:Thanks for your advice. In the process of fire extinguishing, there is mainly heat exchange between AFFF and diesel, and there is no chemical reaction between them.

(ii) Why has author taken diesel fuel rather than any other standard fuels?

Reply:Thanks for your advice. According to GB27897-2011, the fuel used in this manuscript is diesel fuel.

(iii) Must be included few literatures from burning behavior of diesel pool fire inside compartment.

Reply:Thanks for your advice. Some literatures has been cited in new manuscript to illustrate burning behavior of diesel pool fire.

Results and Discussion:

(iv) Section 3.1, 3.2 & 3.3 are not compared with any similar studies held in past, kindly compare presented results with another.

Reply:Thanks for your advice. The purpose of Section 3.1, 3.2 & 3.3 are aimed to compare the fire extinguishing time and temperature drop rate of foam extinguishing agent under different flow rate, gas-liquid mixing ratio and driving pressure. Meanwhile, due to the different fire extinguishing efficiency of foam extinguishing agent under different technical parameters of fire extinguishing system, it is impossible to compare the performance of different fire extinguishing agents. Hence it, the manuscript only compares the experimental results of AFFF under different fire extinguishing technical parameters.

(v) In section 3.4, Equations 1, 2, 3 & 4 are not properly cited i.e., from where it has been taken?

Reply:Thanks for your advice. Equations 1, 2, 3 & 4 are cited from Fundamentals of fire phenomena, and the reference has been added in new manuscript.

(vi) While doing fire suppression modelling of AFFF in diesel pool fire, kindly the quantity of diesel fuel and AFFF along with how reactions are being at mixed gas layer?

Reply:Thanks for your advice. During the fire extinguishing process, the evaporated foam liquid will dilute the concentration of combustible gas in the gas mixing layer, thereby reducing the burning rate, which is also one of the fire extinguishing mechanism of foam fire extinguishing agents.
